# A Closer Look at Few-shot Classification

**Wei-Yu Chen**
Carnegie Mellon University
`weiyuc@andrew.cmu.edu`

**Yen-Cheng Liu & Zsolt Kira**
Georgia Tech
`{ycliu,zkira}@gatech.edu`

**Yu-Chiang Frank Wang**
National Taiwan University
`ycwang@ntu.edu.tw`

**Jia-Bin Huang**
Virginia Tech
`jbhuang@vt.edu`

## Abstract

Few-shot classification aims to learn a classifier to recognize unseen classes during training with limited labeled examples. While significant progress has been made, the growing complexity of network designs, meta-learning algorithms, and differences in implementation details make a fair comparison difficult. In this paper, we present 1) a consistent comparative analysis of several representative few-shot classification algorithms, with results showing that deeper backbones significantly reduce the performance differences among methods on datasets with limited domain differences, 2) a modified baseline method that surprisingly achieves competitive performance when compared with the state-of-the-art on both the *mini*-ImageNet and the CUB datasets, and 3) a new experimental setting for evaluating the cross-domain generalization ability for few-shot classification algorithms. Our results reveal that reducing intra-class variation is an important factor when the feature backbone is shallow, but not as critical when using deeper backbones. In a realistic cross-domain evaluation setting, we show that a baseline method with a standard fine-tuning practice compares favorably against other state-of-the-art few-shot learning algorithms.

## 1 Introduction

Deep learning models have achieved state-of-the-art performance on visual recognition tasks such as image classification. The strong performance, however, heavily relies on training a network with abundant labeled instances with diverse visual variations (e.g., thousands of examples for each new class even with pre-training on large-scale dataset with base classes). The human annotation cost as well as the scarcity of data in some classes (e.g., rare species) significantly limit the applicability of current vision systems to learn new visual concepts efficiently. In contrast, the human visual systems can recognize new classes with extremely few labeled examples. It is thus of great interest to learn to generalize to new classes with a limited amount of labeled examples for each novel class.

The problem of learning to generalize to unseen classes during training, known as *few-shot classification*, has attracted considerable attention Vinyals et al. (2016); Snell et al. (2017); Finn et al. (2017); Ravi & Larochelle (2017); Sung et al. (2018); Garcia & Bruna (2018); Qi et al. (2018). One promising direction to few-shot classification is the meta-learning paradigm where transferable knowledge is extracted and propagated from a collection of tasks to prevent overfitting and improve generalization. Examples include model initialization based methods Ravi & Larochelle (2017); Finn et al. (2017), metric learning methods Vinyals et al. (2016); Snell et al. (2017); Sung et al. (2018), and hallucination based methods Antoniou et al. (2018); Hariharan & Girshick (2017); Wang et al. (2018). Another line of work Gidaris & Komodakis (2018); Qi et al. (2018) also demonstrates promising results by directly predicting the weights of the classifiers for novel classes.

**Limitations.** While many few-shot classification algorithms have reported improved performance over the state-of-the-art, there are two main challenges that prevent us from making a fair comparison and measuring the actual progress. First, the discrepancy of the implementation details among multiple few-shot learning algorithms obscures the relative performance gain. The performance of

baseline approaches can also be significantly under-estimated (e.g., training without data augmentation). Second, while the current evaluation focuses on recognizing novel class with limited training examples, these novel classes are sampled from *the same* dataset. The lack of domain shift between the base and novel classes makes the evaluation scenarios unrealistic.

**Our work.** In this paper, we present a detailed empirical study to shed new light on the few-shot classification problem. First, we conduct consistent comparative experiments to compare several representative few-shot classification methods on common ground. Our results show that using a deep backbone shrinks the performance gap between different methods in the setting of limited domain differences between base and novel classes. Second, by replacing the linear classifier with a distance-based classifier as used in Gidaris & Komodakis (2018); Qi et al. (2018), the baseline method is surprisingly competitive to current state-of-art meta-learning algorithms. Third, we introduce a practical evaluation setting where there exists domain shift between base and novel classes (e.g., sampling base classes from generic object categories and novel classes from fine-grained categories). Our results show that sophisticated few-shot learning algorithms do not provide performance improvement over the baseline under this setting. Through making the source code and model implementations with a consistent evaluation setting publicly available, we hope to foster future progress in the field.[1]

**Our contributions.**

1. We provide a unified testbed for several different few-shot classification algorithms for a fair comparison. Our empirical evaluation results reveal that the use of a shallow backbone commonly used in existing work leads to favorable results for methods that explicitly reduce intra-class variation. Increasing the model capacity of the feature backbone reduces the performance gap between different methods when domain differences are limited.

2. We show that a baseline method with a distance-based classifier surprisingly achieves competitive performance with the state-of-the-art meta-learning methods on both *mini*-ImageNet and CUB datasets.

3. We investigate a practical evaluation setting where base and novel classes are sampled from *different* domains. We show that current few-shot classification algorithms fail to address such domain shifts and are inferior even to the baseline method, highlighting the importance of learning to adapt to domain differences in few-shot learning.

## 2 RELATED WORK

Given abundant training examples for the base classes, few-shot learning algorithms aim to learn to recognizing novel classes with a limited amount of labeled examples. Much efforts have been devoted to overcome the data efficiency issue. In the following, we discuss representative few-shot learning algorithms organized into three main categories: initialization based, metric learning based, and hallucination based methods.

**Initialization based methods** tackle the few-shot learning problem by "learning to fine-tune". One approach aims to learn *good model initialization* (i.e., the parameters of a network) so that the classifiers for novel classes can be learned with a limited number of labeled examples and a small number of gradient update steps Finn et al. (2017; 2018); Nichol & Schulman (2018); Rusu et al. (2019). Another line of work focuses on *learning an optimizer*. Examples include the LSTM-based meta-learner for replacing the stochastic gradient decent optimizer Ravi & Larochelle (2017) and the weight-update mechanism with an external memory Munkhdalai & Yu (2017). While these initialization based methods are capable of achieving rapid adaption with a limited number of training examples for novel classes, our experiments show that these methods have difficulty in handling domain shifts between base and novel classes.

**Distance metric learning based methods** address the few-shot classification problem by "learning to compare". The intuition is that if a model can determine the similarity of two images, it can classify an unseen input image with the labeled instances Koch et al. (2015). To learn a sophisticated comparison models, meta-learning based methods make their prediction conditioned on distance or

---

[1] https://github.com/wyharveychen/CloserLookFewShot

metric to few labeled instances during the training process. Examples of distance metrics include cosine similarity Vinyals et al. (2016), Euclidean distance to class-mean representation Snell et al. (2017), CNN-based relation module Sung et al. (2018), ridge regression Bertinetto et al. (2019), and graph neural network Garcia & Bruna (2018). In this paper, we compare the performance of three distance metric learning methods. Our results show that a simple baseline method with a distance-based classifier (without training over a collection of tasks/episodes as in meta-learning) achieves competitive performance with respect to other sophisticated algorithms.

Besides meta-learning methods, both Gidaris & Komodakis (2018) and Qi et al. (2018) develop a similar method to our Baseline++ (described later in Section 3.2). The method in Gidaris & Komodakis (2018) learns a weight generator to predict the novel class classifier using an attention-based mechanism (cosine similarity), and the Qi et al. (2018) directly use novel class features as their weights. Our Baseline++ can be viewed as a simplified architecture of these methods. Our focus, however, is to show that simply reducing intra-class variation in a baseline method using the base class data leads to competitive performance.

**Hallucination based methods** directly deal with data deficiency by "learning to augment". This class of methods learns a generator from data in the base classes and use the learned generator to hallucinate new novel class data for data augmentation. One type of generator aims at transferring appearance variations exhibited in the base classes. These generators either transfer variance in base class data to novel classes Hariharan & Girshick (2017), or use GAN models Antoniou et al. (2018) to transfer the style. Another type of generators does not explicitly specify what to transfer, but directly integrate the generator into a meta-learning algorithm for improving the classification accuracy Wang et al. (2018). Since hallucination based methods often work with other few-shot methods together (e.g. use hallucination based and metric learning based methods together) and lead to complicated comparison, we do not include these methods in our comparative study and leave it for future work.

**Domain adaptation** techniques aim to reduce the domain shifts between source and target domain Pan et al. (2010); Ganin & Lempitsky (2015), as well as novel tasks in a different domain Hsu et al. (2018). Similar to domain adaptation, we also investigate the impact of domain difference on few-shot classification algorithms in Section 4.5. In contrast to most domain adaptation problems where a large amount of data is available in the target domain (either labeled or unlabeled), our problem setting differs because we only have very few examples in the new domain. Very recently, the method in Dong & Xing (2018) addresses the one-shot novel category domain adaptation problem, where in the testing stage both the domain *and* the category to classify are changed. Similarly, our work highlights the limitations of existing few-shot classification algorithms problem in handling domain shift. To put these problem settings in context, we provided a detailed comparison of setting difference in the appendix A1.

## 3 OVERVIEW OF FEW-SHOT CLASSIFICATION ALGORITHMS

In this section, we first outline the details of the baseline model (Section 3.1) and its variant (Section 3.2), followed by describing representative meta-learning algorithms (Section 3.3) studied in our experiments. Given abundant base class labeled data $\mathbf{X}_b$ and a small amount of novel class labeled data $\mathbf{X}_n$, the goal of few-shot classification algorithms is to train classifiers for novel classes (unseen during training) with few labeled examples.

### 3.1 BASELINE

Our baseline model follows the standard transfer learning procedure of network pre-training and fine-tuning. Figure 1 illustrates the overall procedure.

**Training stage.** We train a feature extractor $f_\theta$ (parametrized by the network parameters $\theta$) and the classifier $C(\cdot|\mathbf{W}_b)$ (parametrized by the weight matrix $\mathbf{W}_b \in \mathbb{R}^{d \times c}$) from scratch by minimizing a standard cross-entropy classification loss $L_{\text{pred}}$ using the training examples in the base classes $\mathbf{x}_i \in \mathbf{X}_b$. Here, we denote the dimension of the encoded feature as $d$ and the number of output classes as $c$. The classifier $C(.|\mathbf{W}_b)$ consists of a linear layer $\mathbf{W}_b^\top f_\theta(\mathbf{x}_i)$ followed by a softmax function $\sigma$.

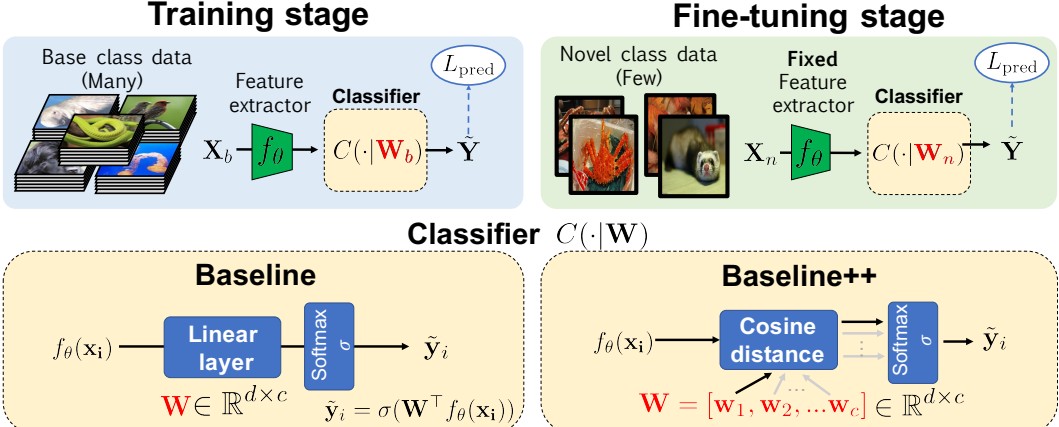

Figure 1: **Baseline and Baseline++ few-shot classification methods.** Both the baseline and baseline++ method train a feature extractor $f_\theta$ and classifier $C(.|\mathbf{W}_b)$ with base class data in the training stage In the fine-tuning stage, we fix the network parameters $\theta$ in the feature extractor $f_\theta$ and train a new classifier $C(.|\mathbf{W}_n)$ with the given labeled examples in novel classes. The baseline++ method differs from the baseline model in the use of cosine distances between the input feature and the weight vector for each class that aims to reduce intra-class variations.

**Fine-tuning stage.** To adapt the model to recognize novel classes in the fine-tuning stage, we fix the pre-trained network parameter $\theta$ in our feature extractor $f_\theta$ and train a new classifier $C(.|\mathbf{W}_n)$ (parametrized by the weight matrix $\mathbf{W}_n$) by minimizing $L_{\text{pred}}$ using the few labeled of examples (i.e., the support set) in the novel classes $\mathbf{X}_n$.

## 3.2 BASELINE++

In addition to the baseline model, we also implement a variant of the baseline model, denoted as Baseline++, which explicitly reduces intra-class variation among features during training. The importance of reducing intra-class variations of features has been highlighted in deep metric learning Hu et al. (2015) and few-shot classification methods Gidaris & Komodakis (2018).

The training procedure of Baseline++ is the same as the original Baseline model except for the classifier design. As shown in Figure 1, we still have a weight matrix $\mathbf{W}_b \in \mathbb{R}^{d \times c}$ of the classifier in the training stage and a $\mathbf{W}_n$ in the fine-tuning stage in Baseline++. The classifier design, however, is different from the linear classifier used in the Baseline. Take the weight matrix $\mathbf{W}_b$ as an example. We can write the weight matrix $\mathbf{W}_b$ as $[\mathbf{w}_1, \mathbf{w}_2, ...\mathbf{w}_c]$, where each class has a $d$-dimensional weight vector. In the training stage, for an input feature $f_\theta(\mathbf{x}_i)$ where $\mathbf{x}_i \in \mathbf{X}_b$, we compute its cosine similarity to each weight vector $[\mathbf{w}_1, \cdots, \mathbf{w}_c]$ and obtain the similarity scores $[s_{i,1}, s_{i,2}, \cdots, s_{i,c}]$ for all classes, where $s_{i,j} = f_\theta(\mathbf{x}_i)^\top \mathbf{w}_j / \|f_\theta(\mathbf{x}_i)\| \|\mathbf{w}_j\|$. We can then obtain the prediction probability for each class by normalizing these similarity scores with a softmax function. Here, the classifier makes a prediction based on the cosine distance between the input feature and the learned weight vectors representing each class. Consequently, training the model with this distance-based classifier explicitly reduce intra-class variations. Intuitively, the learned weight vectors $[\mathbf{w}_1, \cdots, \mathbf{w}_c]$ can be interpreted as prototypes (similar to Snell et al. (2017); Vinyals et al. (2016)) for each class and the classification is based on the distance of the input feature to these learned prototypes. The softmax function prevents the learned weight vectors collapsing to zeros.

We clarify that the network design in Baseline++ is *not* our contribution. The concept of distance-based classification has been extensively studied in Mensink et al. (2012) and recently has been revisited in the few-shot classification setting Gidaris & Komodakis (2018); Qi et al. (2018).

## 3.3 META-LEARNING ALGORITHMS

Here we describe the formulations of meta-learning methods used in our study. We consider three distance metric learning based methods (MatchingNet Vinyals et al. (2016), ProtoNet Snell et al.

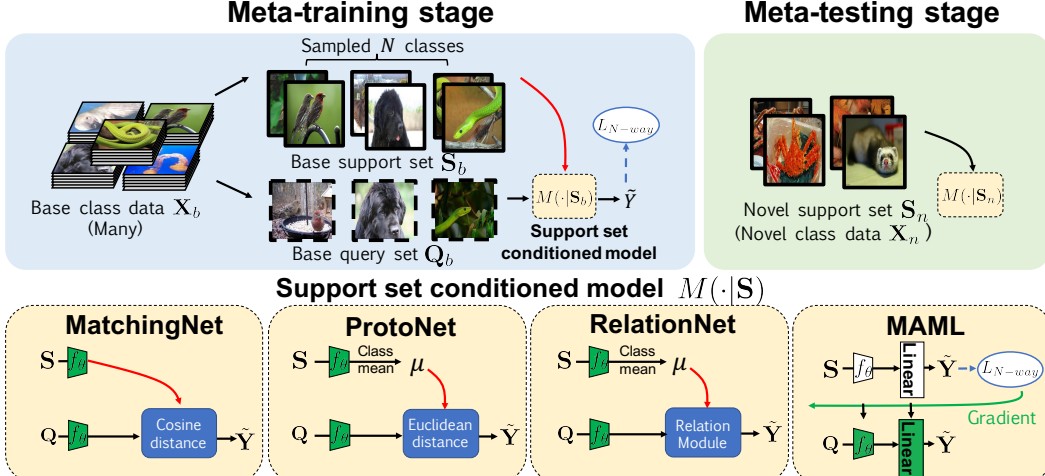

Figure 2: **Meta-learning few-shot classification algorithms.** The meta-learning classifier $M(\cdot|\mathbf{S})$ is conditioned on the support set $\mathbf{S}$. (*Top*) In the meta-train stage, the support set $\mathbf{S}_b$ and the query set $\mathbf{Q}_b$ are first sampled from random $N$ classes, and then train the parameters in $M(.|\mathbf{S}_b)$ to minimize the $N$-way prediction loss $L_{N-\text{way}}$. In the meta-testing stage, the adapted classifier $M(.|\mathbf{S}_n)$ can predict novel classes with the support set in the novel classes $\mathbf{S}_n$. (*Bottom*) The design of $M(\cdot|\mathbf{S})$ in different meta-learning algorithms.

(2017), and RelationNet Sung et al. (2018)) and one initialization based method (MAML Finn et al. (2017)). While meta-learning is not a clearly defined, Vinyals et al. (2016) considers a few-shot classification method as meta-learning if the prediction is conditioned on a small support set $\mathbf{S}$, because it makes the training procedure explicitly learn to learn from a given small support set.

As shown in Figure 2, meta-learning algorithms consist of a meta-training and a meta-testing stage. In the meta-training stage, the algorithm first randomly select $N$ classes, and sample small base support set $\mathbf{S}_b$ and a base query set $\mathbf{Q}_b$ from data samples within these classes. The objective is to train a classification model $M$ that minimizes $N$-way prediction loss $L_{N-\text{way}}$ of the samples in the query set $\mathbf{Q}_b$. Here, the classifier $M$ is conditioned on provided support set $\mathbf{S}_b$. By making prediction conditioned on the given support set, a meta-learning method can learn how to learn from limited labeled data through training from a collection of tasks (episodes). In the meta-testing stage, all novel class data $\mathbf{X}_n$ are considered as the support set for novel classes $\mathbf{S}_n$, and the classification model $M$ can be adapted to predict novel classes with the new support set $\mathbf{S}_n$.

Different meta-learning methods differ in their strategies to make prediction conditioned on support set (see Figure 2). For both MatchingNet Vinyals et al. (2016) and ProtoNet Snell et al. (2017), the prediction of the examples in a query set $\mathbf{Q}$ is based on comparing the distance between the query feature and the support feature from each class. MatchingNet compares cosine distance between the query feature and each support feature, and computes average cosine distance for each class, while ProtoNet compares the Euclidean distance between query features and the class mean of support features. RelationNet Sung et al. (2018) shares a similar idea, but it replaces distance with a learnable relation module. The MAML method Finn et al. (2017) is an initialization based meta-learning algorithm, where each support set is used to adapt the initial model parameters using few gradient updates. As different support sets have different gradient updates, the adapted model is conditioned on the support set. Note that when the query set instances are predicted by the adapted model in the meta-training stage, the loss of the query set is used to update the initial model, not the adapted model.

# 4 EXPERIMENTAL RESULTS

## 4.1 EXPERIMENTAL SETUP

**Datasets and scenarios.** We address the few-shot classification problem under three scenarios: 1) generic object recognition, 2) fine-grained image classification, and 3) cross-domain adaptation.

For object recognition, we use the *mini*-ImageNet dataset commonly used in evaluating few-shot classification algorithms. The *mini*-ImageNet dataset consists of a subset of 100 classes from the ImageNet dataset Deng et al. (2009) and contains 600 images for each class. The dataset was first proposed by Vinyals et al. (2016), but recent works use the follow-up setting provided by Ravi & Larochelle (2017), which is composed of randomly selected 64 base, 16 validation, and 20 novel classes.

For fine-grained classification, we use CUB-200-2011 dataset Wah et al. (2011) (referred to as the CUB hereafter). The CUB dataset contains 200 classes and 11,788 images in total. Following the evaluation protocol of Hilliard et al. (2018), we randomly split the dataset into 100 base, 50 validation, and 50 novel classes.

For the cross-domain scenario (*mini*-ImageNet →CUB), we use *mini*-ImageNet as our base class and the 50 validation and 50 novel class from CUB. Evaluating the cross-domain scenario allows us to understand the effects of domain shifts to existing few-shot classification approaches.

**Implementation details.**   In the training stage for the Baseline and the Baseline++ methods, we train 400 epochs with a batch size of 16. In the meta-training stage for meta-learning methods, we train 60,000 episodes for 1-shot and 40,000 episodes for 5-shot tasks. We use the validation set to select the training episodes with the best accuracy.[2] In each episode, we sample $N$ classes to form $N$-way classification ($N$ is 5 in both meta-training and meta-testing stages unless otherwise mentioned). For each class, we pick $k$ labeled instances as our support set and 16 instances for the query set for a $k$-shot task.

In the fine-tuning or meta-testing stage for all methods, we average the results over 600 experiments. In each experiment, we randomly sample 5 classes from novel classes, and in each class, we also pick $k$ instances for the support set and 16 for the query set. For Baseline and Baseline++, we use the entire support set to train a new classifier for 100 iterations with a batch size of 4. For meta-learning methods, we obtain the classification model conditioned on the support set as in Section 3.3.

All methods are trained from scratch and use the Adam optimizer with initial learning rate $10^{-3}$. We apply standard data augmentation including random crop, left-right flip, and color jitter in both the training or meta-training stage. Some implementation details have been adjusted individually for each method. For Baseline++, we multiply the cosine similarity by a constant scalar 2 to adjust original value range [-1,1] to be more appropriate for subsequent softmax layer. For MatchingNet, we use an FCE classification layer without fine-tuning in all experiments and also multiply cosine similarity by a constant scalar. For RelationNet, we replace the L2 norm with a softmax layer to expedite training. For MAML, we use a first-order approximation in the gradient for memory efficiency. The approximation has been shown in the original paper and in our appendix to have nearly identical performance as the full version. We choose the first-order approximation for its efficiency.

## 4.2   EVALUATION USING THE STANDARD SETTING

We now conduct experiments on the most common setting in few-shot classification, 1-shot and 5-shot classification, i.e., 1 or 5 labeled instances are available from each novel class. We use a four-layer convolution backbone (Conv-4) with an input size of 84x84 as in Snell et al. (2017) and perform 5-way classification for only novel classes during the fine-tuning or meta-testing stage.

To validate the correctness of our implementation, we first compare our results to the reported numbers for the *mini*-ImageNet dataset in Table 1. Note that we have a ProtoNet[#], as we use 5-way classification in the meta-training and meta-testing stages for all meta-learning methods as mentioned in Section 4.1; however, the official reported results from ProtoNet uses 30-way for one shot and 20-way for five shot in the meta-training stage in spite of using 5-way in the meta-testing stage. We report this result for completeness.

From Table 1, we can observe that all of our re-implementation for meta-learning methods do not fall more than 2% behind reported performance. These minor differences can be attributed to our

---

[2]For example, the exact episodes for experiments on the mini-ImageNet in the 5-shot setting with a four-layer ConvNet are: ProtoNet: 24,600; MatchingNet: 35,300; RelationNet: 37,100; MAML: 36,700.

[3]Reported results are from Ravi & Larochelle (2017)

Table 1: **Validating our re-implementation.** We validate our few-shot classification implementation on the *mini*-ImageNet dataset using a Conv-4 backbone. We report the mean of 600 randomly generated test episodes as well as the 95% confidence intervals. Our reproduced results to all few-shot methods do not fall behind by more than 2% to the reported results in the literature. We attribute the slight discrepancy to different random seeds and minor implementation differences in each method. "Baseline*" denotes the results without applying data augmentation during training. ProtoNet[#] indicates performing 30-way classification in 1-shot and 20-way in 5-shot during the meta-training stage.

| Method | 1-shot | | 5-shot | |
| --- | --- | --- | --- | --- |
| | Reported | Ours | Reported | Ours |
| **Baseline** | - | $42.11 \pm 0.71$ | - | $62.53 \pm 0.69$ |
| **Baseline***[3] | $41.08 \pm 0.70$ | $36.35 \pm 0.64$ | $51.04 \pm 0.65$ | $54.50 \pm 0.66$ |
| **MatchingNet**[3] Vinyals et al. (2016) | $43.56 \pm 0.84$ | $48.14 \pm 0.78$ | $55.31 \pm 0.73$ | $63.48 \pm 0.66$ |
| **ProtoNet** | - | $44.42 \pm 0.84$ | - | $64.24 \pm 0.72$ |
| **ProtoNet**[#] Snell et al. (2017) | $49.42 \pm 0.78$ | $47.74 \pm 0.84$ | $68.20 \pm 0.66$ | $66.68 \pm 0.68$ |
| **MAML** Finn et al. (2017) | $48.07 \pm 1.75$ | $46.47 \pm 0.82$ | $63.15 \pm 0.91$ | $62.71 \pm 0.71$ |
| **RelationNet** Sung et al. (2018) | $50.44 \pm 0.82$ | $49.31 \pm 0.85$ | $65.32 \pm 0.70$ | $66.60 \pm 0.69$ |

Table 2: **Few-shot classification results for both the *mini*-ImageNet and *CUB* datasets.** The Baseline++ consistently improves the Baseline model by a large margin and is competitive with the state-of-the-art meta-learning methods. All experiments are from 5-way classification with a Conv-4 backbone and data augmentation.

| Method | CUB | | *mini*-ImageNet | |
| --- | --- | --- | --- | --- |
| | 1-shot | 5-shot | 1-shot | 5-shot |
| **Baseline** | $47.12 \pm 0.74$ | $64.16 \pm 0.71$ | $42.11 \pm 0.71$ | $62.53 \pm 0.69$ |
| **Baseline++** | $60.53 \pm 0.83$ | $79.34 \pm 0.61$ | $48.24 \pm 0.75$ | $66.43 \pm 0.63$ |
| **MatchingNet** Vinyals et al. (2016) | $61.16 \pm 0.89$ | $72.86 \pm 0.70$ | $48.14 \pm 0.78$ | $63.48 \pm 0.66$ |
| **ProtoNet** Snell et al. (2017) | $51.31 \pm 0.91$ | $70.77 \pm 0.69$ | $44.42 \pm 0.84$ | $64.24 \pm 0.72$ |
| **MAML** Finn et al. (2017) | $55.92 \pm 0.95$ | $72.09 \pm 0.76$ | $46.47 \pm 0.82$ | $62.71 \pm 0.71$ |
| **RelationNet** Sung et al. (2018) | $62.45 \pm 0.98$ | $76.11 \pm 0.69$ | $49.31 \pm 0.85$ | $66.60 \pm 0.69$ |

modifications of some implementation details to ensure a fair comparison among all methods, such as using the same optimizer for all methods.

Moreover, our implementation of existing work also improves the performance of some of the methods. For example, our results show that the Baseline approach under 5-shot setting can be improved by a large margin since previous implementations of the Baseline do not include data augmentation in their training stage, thereby leads to over-fitting. While our Baseline* is not as good as reported in 1-shot, our Baseline with augmentation still improves on it, and could be even higher if our reproduced Baseline* matches the reported statistics. In either case, **the performance of the Baseline method is severely underestimated**. We also improve the results of MatchingNet by adjusting the input score to the softmax layer to a more appropriate range as stated in Section 4.1. On the other hand, while ProtoNet[#] is not as good as ProtoNet, as mentioned in the original paper a more challenging setting in the meta-training stage leads to better accuracy. We choose to use a consistent 5-way classification setting in subsequent experiments to have a fair comparison to other methods. This issue can be resolved by using a deeper backbone as shown in Section 4.3.

After validating our re-implementation, we now report the accuracy in Table 2. Besides additionally reporting results on the CUB dataset, we also compare Baseline++ to other methods. Here, we find that Baseline++ improves the Baseline by a large margin and becomes competitive even when compared with other meta-learning methods. The results demonstrate that **reducing intra-class variation is an important factor in the current few-shot classification problem setting**.

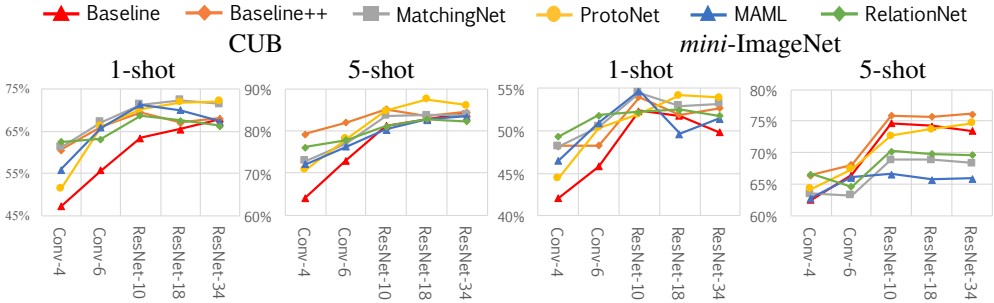

Figure 3: **Few-shot classification accuracy vs. backbone depth**. In the CUB dataset, gaps among different methods diminish as the backbone gets deeper. In *mini*-ImageNet 5-shot, some meta-learning methods are even beaten by Baseline with a deeper backbone. (Please refer to Figure A3 and Table A5 for larger figure and detailed statistics.)

However, note that our current setting only uses a 4-layer backbone, while a deeper backbone can inherently reduce intra-class variation. Thus, we conduct experiments to investigate the effects of backbone depth in the next section.

### 4.3 EFFECT OF INCREASING THE NETWORK DEPTH

In this section, we change the depth of the feature backbone to reduce intra-class variation for all methods. See appendix for statistics on how network depth correlates with intra-class variation. Starting from Conv-4, we gradually increase the feature backbone to Conv-6, ResNet-10, 18 and 34, where Conv-6 have two additional convolution blocks without pooling after Conv-4. ResNet-18 and 34 are the same as described in He et al. (2016) with an input size of 224×224, while ResNet-10 is a simplified version of ResNet-18 where only one residual building block is used in each layer. The statistics of this experiment would also be helpful to other works to make a fair comparison under different feature backbones.

Results of the CUB dataset shows a clearer tendency in Figure 3. As the backbone gets deeper, the gap among different methods drastically reduces. Another observation is how ProtoNet improves rapidly as the backbone gets deeper. While using a consistent 5-way classification as discussed in Section 4.2 degrades the accuracy of ProtoNet with Conv-4, it works well with a deeper backbone. Thus, the two observations above demonstrate that **in the CUB dataset, the gap among existing methods would be reduced if their intra-class variation are all reduced by a deeper backbone.**

However, the result of *mini*-ImageNet in Figure 3 is much more complicated. In the 5-shot setting, both Baseline and Baseline++ achieve good performance with a deeper backbone, but some meta-learning methods become worse relative to them. Thus, other than intra-class variation, we can assume that the dataset is also important in few-shot classification. One difference between CUB and *mini*-ImageNet is their domain difference in base and novel classes since classes in *mini*-ImageNet have a larger divergence than CUB in a word-net hierarchy Miller (1995). To better understand the effect, below we discuss how domain differences between base and novel classes impact few-shot classification results.

### 4.4 EFFECT OF DOMAIN DIFFERENCES BETWEEN BASE AND NOVEL CLASSES

To further dig into the issue of domain difference, we design scenarios that provide such domain shifts. Besides the fine-grained classification and object recognition scenarios, we propose a new *cross-domain scenario*: *mini*-ImageNet →CUB as mentioned in  Section 4.1. We believe that this is practical scenario since collecting images from a general class may be relatively easy (e.g. due to increased availability) but collecting images from fine-grained classes might be more difficult.

We conduct the experiments with a ResNet-18 feature backbone. As shown in Table 3, the Baseline outperforms all meta-learning methods under this scenario. While meta-learning methods learn to learn from the support set during the meta-training stage, they are not able to adapt to novel classes that are too different since all of the base support sets are within the same dataset. A similar concept is also mentioned in Vinyals et al. (2016). In contrast, the Baseline simply replaces and trains a new classifier based on the few given novel class data, which allows it to quickly adapt to a novel

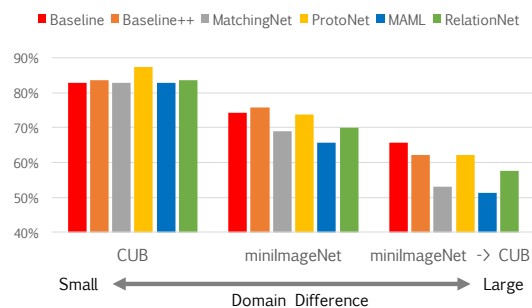

|  | *mini*-ImageNet →CUB |
|---|---|
| **Baseline** | **65.57±0.70** |
| **Baseline++** | 62.04±0.76 |
| **MatchingNet** | 53.07±0.74 |
| **ProtoNet** | 62.02±0.70 |
| **MAML** | 51.34±0.72 |
| **RelationNet** | 57.71±0.73 |

Table 3: **5-shot accuracy under the cross-domain scenario with a ResNet-18 backbone.** Baseline outperforms all other methods under this scenario.

Figure 4: **5-shot accuracy in different scenarios with a ResNet-18 backbone.** The Baseline model performs relative well with larger domain differences.

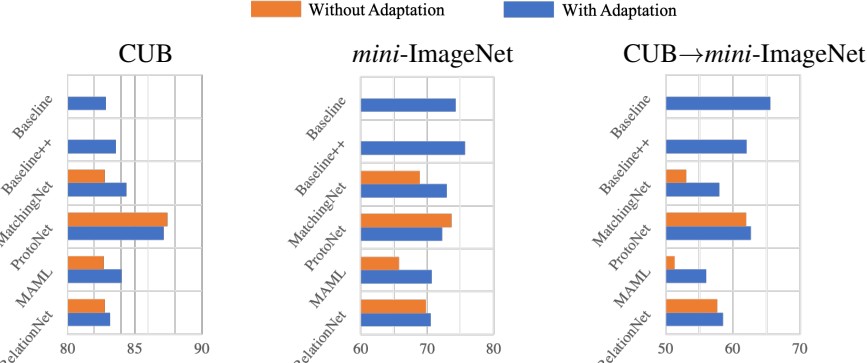

Figure 5: **Meta-learning methods with further adaptation steps.** Further adaptation improves MatchingNet and MAML, but has less improvement to RelationNet, and could instead harm ProtoNet under the scenarios with little domain differences. All statistics are for 5-shot accuracy with ResNet-18 backbone. Note that different methods use different further adaptation strategies.

class and is less affected by domain shift between the source and target domains. The Baseline also performs better than the Baseline++ method, possibly because additionally reducing intra-class variation compromises adaptability. In Figure 4, we can further observe how Baseline accuracy becomes relatively higher as the domain difference gets larger. That is, **as the domain difference grows larger, the adaptation based on a few novel class instances becomes more important.**

## 4.5 EFFECT OF FURTHER ADAPTATION

To further adapt meta-learning methods as in the Baseline method, an intuitive way is to fix the features and train a new softmax classifier. We apply this simple adaptation scheme to MatchingNet and ProtoNet. For MAML, it is not feasible to fix the feature as it is an initialization method. In contrast, since it updates the model with the support set for only a few iterations, we can adapt further by updating for as many iterations as is required to train a new classification layer, which is 100 updates as mentioned in Section 4.1. For RelationNet, the features are convolution maps rather than the feature vectors, so we are not able to replace it with a softmax. As an alternative, we randomly split the few training data in novel class into 3 support and 2 query data to finetune the relation module for 100 epochs.

The results of further adaptation are shown in Figure 5; we can observe that the performance of MatchingNet and MAML improves significantly after further adaptation, particularly in the *mini*-ImageNet →CUB scenario. The results demonstrate that lack of adaptation is the reason they fall behind the Baseline. However, changing the setting in the meta-testing stage can lead to inconsistency with the meta-training stage. The ProtoNet result shows that performance can degrade in sce-

narios with less domain difference. Thus, we believe that *learning how to adapt* in the meta-training stage is important future direction. In summary, as domain differences are likely to exist in many real-world applications, we consider that **learning to learn adaptation in the meta-training stage would be an important direction for future meta-learning research in few-shot classification.**

## 5 CONCLUSIONS

In this paper, we have investigated the limits of the standard evaluation setting for few-shot classification. Through comparing methods on a common ground, our results show that the Baseline++ model is competitive to state of art under standard conditions, and the Baseline model achieves competitive performance with recent state-of-the-art meta-learning algorithms on both CUB and *mini*-ImageNet benchmark datasets when using a deeper feature backbone. Surprisingly, the Baseline compares favorably against all the evaluated meta-learning algorithms under a realistic scenario where there exists domain shift between the base and novel classes. By making our source code publicly available, we believe that community can benefit from the consistent comparative experiments and move forward to tackle the challenge of potential domain shifts in the context of few-shot learning.

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

## APPENDIX

### A1 RELATIONSHIP BETWEEN DOMAIN ADAPTATION AND FEW-SHOT CLASSIFICATION

As mentioned in Section 2, here we discuss the relationship between *domain adaptation* and *few-shot classification* to clarify different experimental settings. As shown in Table A1, in general, domain adaptation aims at adapting source dataset knowledge to the *same* class in target dataset. On the other hand, the goal of few-shot classification is to learn from base classes to classify *novel* classes in the same dataset.

Several recent work tackle the problem at the intersection of the two fields of study. For example, cross-task domain adaptation Hsu et al. (2018) also discuss novel classes in the target dataset. In contrast, while Motiian et al. (2017) has "few-shot" in the title, their evaluation setting focuses on classifying the same class in the target dataset.

If base and novel classes are both drawn from the same dataset, minor domain shift exists between the base and novel classes, as we demonstrated in Section 4.4. To highlight the impact of domain shift, we further propose the *mini*-ImageNet →CUB setting. The domain shift in few-shot classification is also discussed in Dong & Xing (2018).

Table A1: **Relationship between domain adaptation and few-shot classification.** The two field-of-studies have overlapping in the development. Notation "*" indicates minor domain shifts exist between base and novel classes.

|  | Domain shift | Source to target dataset | Base to novel class |
|---|---|---|---|
| Domain adaptation Motiian et al. (2017) | V | V | - |
| Cross-task domain adaptation Hsu et al. (2018) | V | V | V |
| Few-shot classification Ours (CUB, *mini*-ImageNet ) | * | - | V |
| Cross-domain few-shot Ours (*mini*-ImageNet →CUB) Dong & Xing (2018) | V | V | V |

### A2 TERMINOLOGY DIFFERENCE

Different meta-learning works use different terminology in their works. We highlight their differences in appendix Table A2 to clarify the inconsistency.

Table A2: **Different terminology used in other works.** Notation "-" indicates the term is the same as in this paper.

| Our terms | MatchingNet Vinyals et al. | ProtoNet Snell et al. | MAML Finn et al. | Meta-learn LSTM Ravi & Larochelle | Imaginary Wang et al. |
|---|---|---|---|---|---|
| meta-training stage | training | training | - | - | - |
| meta-testing stage | test | test | - | - | - |
| base class | training set | training set | task | meta-training set | - |
| novel class | test set | test set | new task | meta-testing set | - |
| support set | - | - | sample | training dataset | training data |
| query set | batch | - | test time sample | test dataset | test data |

## A3 ADDITIONAL RESULTS ON OMNIGLOT AND OMNIGLOT→EMNIST

For completeness, here we also show the results under two additional scenarios in 4) character recognition 5) cross-domain character recognition.

For character recognition, we use the Omniglot dataset Lake et al. (2011) commonly used in evaluating few-shot classification algorithms. Omniglot contains 1,623 characters from 50 languages, and we follow the evaluation protocol of Vinyals et al. (2016) to first augment the classes by rotations in 90, 180, 270 degrees, resulting in 6492 classes. We then follow Snell et al. (2017) to split these classes into 4112 base, 688 validation, and 1692 novel classes. Unlike Snell et al. (2017), our validation classes are only used to monitor the performance during meta-training.

For cross-domain character recognition (Omniglot→EMNIST), we follow the setting of Dong & Xing (2018) to use Omniglot without Latin characters and without rotation augmentation as base classes, so there are 1597 base classes. On the other hand, EMNIST dataset Cohen et al. (2017) contains 10-digits and upper and lower case alphabets in English, so there are 62 classes in total. We split these classes into 31 validation and 31 novel classes, and invert the white-on-black characters to black-on-white as in Omniglot.

We use a Conv-4 backbone with input size 28x28 for both settings. As Omniglot characters are black-and-white, center-aligned and rotation sensitive, we do not use data augmentation in this experiment. To reduce the risk of over-fitting, we use the validation set to select the epoch or episode with the best accuracy for all methods, including baseline and baseline++.[4]

As shown in Table A3, in both Omniglot and Omniglot→EMNIST settings, meta-learning methods outperform baseline and baseline++ in 1-shot. However, all methods reach comparable performance in the 5-shot classification setting. We attribute this to the lack of data augmentation for the baseline and baseline++ methods as they tend to over-fit base classes. When sufficient examples in novel classes are available, the negative impact of over-fitting is reduced.

Table A3: **Few-shot classification results for both the *Omniglot* and *Omniglot*→*EMNIST*.** All experiments are from 5-way classification with a Conv-4 backbone and without data augmentation.

| Method | Omniglot | | Omniglot→EMNIST | |
|---|---|---|---|---|
| | 1-shot | 5-shot | 1-shot | 5-shot |
| **Baseline** | $94.89 \pm 0.45$ | $99.12 \pm 0.13$ | $63.94 \pm 0.87$ | $86.00 \pm 0.59$ |
| **Baseline++** | $95.41 \pm 0.39$ | $99.38 \pm 0.10$ | $64.74 \pm 0.82$ | $87.31 \pm 0.58$ |
| **MatchingNet** | $97.78 \pm 0.30$ | $99.37 \pm 0.11$ | $72.71 \pm 0.79$ | $87.60 \pm 0.56$ |
| **ProtoNet** | $98.01 \pm 0.30$ | $99.15 \pm 0.12$ | $70.43 \pm 0.80$ | $87.04 \pm 0.55$ |
| **MAML** | $98.57 \pm 0.19$ | $99.53 \pm 0.08$ | $72.04 \pm 0.83$ | $88.24 \pm 0.56$ |
| **RelationNet** | $97.22 \pm 0.33$ | $99.30 \pm 0.10$ | $75.55 \pm 0.87$ | $88.94 \pm 0.54$ |

## A4 BASELINE WITH 1-NN CLASSIFIER

Some prior work (Vinyals et al. (2016)) apply a Baseline with 1-NN classifier in the test stage. We include our result as in Table A4. The result shows that using 1-NN classifier has better performance than that of using the softmax classifier in 1-shot setting, but softmax classifier performs better in 5-shot setting. We note that the number here are not directly comparable to results in Vinyals et al. (2016) because we use a different *mini*-ImageNet as in Ravi & Larochelle (2017).

Table A4: **Baseline with softmax and 1-NN classifier in test stage.** We note that we use cosine distance in 1-NN.

| | 1-shot | | 5-shot | |
|---|---|---|---|---|
| | softmax | 1-NN | softmax | 1-NN |
| **Baseline** | $42.11 \pm 0.71$ | $44.18 \pm 0.69$ | $62.53 \pm 0.69$ | $56.68 \pm 0.67$ |
| **Baseline++** | $48.24 \pm 0.75$ | $49.57 \pm 0.73$ | $66.43 \pm 0.63$ | $61.93 \pm 0.65$ |

---

[4]The exact epoch of baseline and baseline++ on Omniglot and Omniglot→EMNIST is 5 epochs

## A5 MAML AND MAML WITH FIRST-ORDER APPROXIMATION

As discussed in Section 4.1, we use first-order approximation MAML to improve memory efficiency in all of our experiments. To demonstrate this design choice does not affect the accuracy, we compare their validation accuracy trends on Omniglot with 5-shot as in Figure A1. We observe that while the full version MAML converge faster, both versions reach similar accuracy in the end.

This phenomena is consistent with the difference of first-order (e.g. gradient descent) and second-order methods (e.g. Newton) in convex optimization problems. Second-order methods converge faster at the cost of memory, but they both converge to similar objective value.

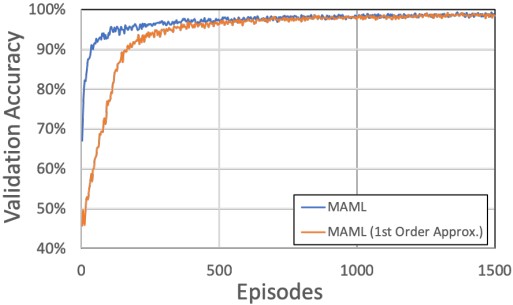

Figure A1: **Validation accuracy trends of MAML and MAML with first order approximation.** Both versions converge to the same validation accuracy. The experimental results are on Omniglot with 5-shot with a Conv-4 backbone.

## A6 INTRA-CLASS VARIATION AND BACKBONE DEPTH

As mentioned in Section 4.3, here we demonstrate decreased intra-class variation as the network depth gets deeper as in Figure A2. We use the Davies-Bouldin index Davies & Bouldin (1979) to measure intra-class variation. The Davies-Bouldin index is a metric to evaluate the tightness in a cluster (or class, in our case). Our results show that both intra-class variation in the base and novel class feature decrease using deeper backbones.

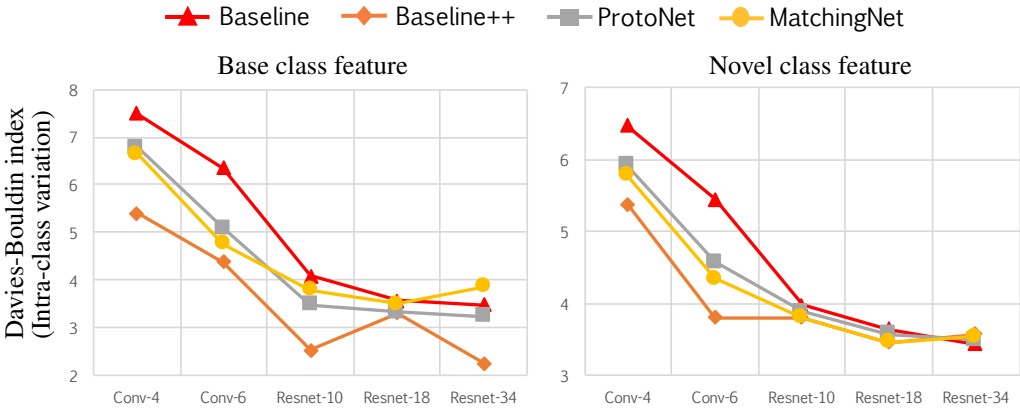

Figure A2: **Intra-class variation decreases as backbone gets deeper.** Here we use Davies-Bouldin index to represent intra-class variation, which is a metric to evaluate the tightness in a cluster (or class, in our case). The statistics are Davies-Bouldin index for all base and novel class feature (extracted by feature extractor learned after training or meta-training stage) for CUB dataset under different backbone.

## A7 DETAILED STATISTICS IN EFFECTS OF INCREASING BACKBONE DEPTH

Here we show a high-resolution version of Figure 3 in Figure A3 and show detailed statistics in Table A5 for easier comparison.

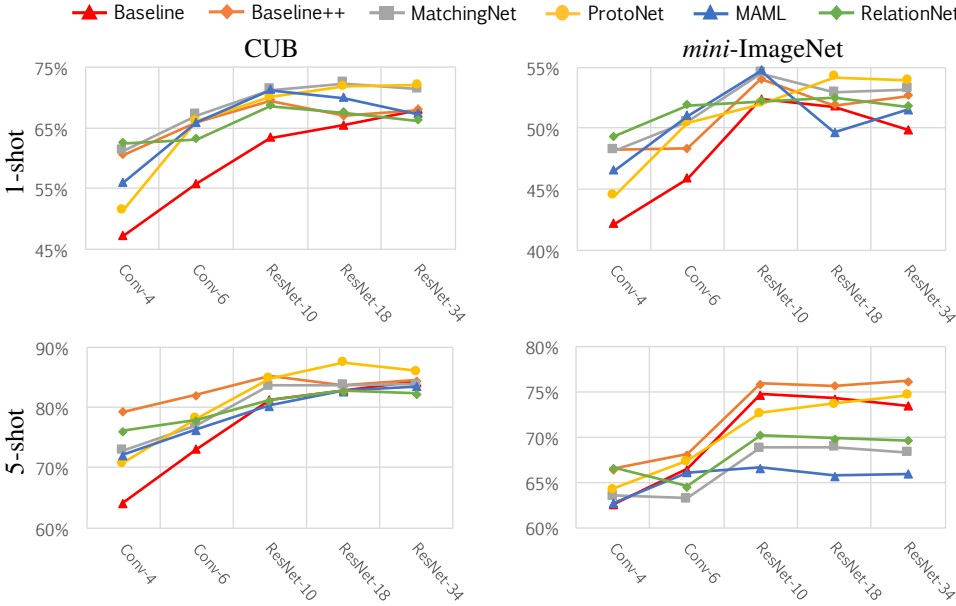

Figure A3: **Few-shot classification accuracy vs. backbone depth**. In the CUB dataset, gaps among different methods diminish as the backbone gets deeper. In *mini*-ImageNet 5-shot, some meta-learning methods are even beaten by Baseline with a deeper backbone.

Table A5: **Detailed statistics in Figure 3.** We put exact value here for reference.

|  |  | Conv-4 | Conv-6 | Resnet-10 | Resnet-18 | Resnet-34 |
|---|---|---|---|---|---|---|
| CUB 1-shot | **Baseline** | 47.12±0.74 | 55.77±0.86 | 63.34±0.91 | 65.51±0.87 | 67.96±0.89 |
|  | **Baseline++** | 60.53±0.83 | 66.00±0.89 | 69.55±0.89 | 67.02±0.90 | 68.00±0.83 |
|  | **MatchingNet** | 61.16±0.89 | 67.16±0.97 | 71.29±0.90 | 72.36±0.90 | 71.44±0.96 |
|  | **ProtoNet** | 51.31±0.91 | 66.07±0.97 | 70.13±0.94 | 71.88±0.91 | 72.03±0.91 |
|  | **MAML** | 55.92±0.95 | 65.91±0.97 | 71.29±0.95 | 69.96±1.01 | 67.28±1.08 |
|  | **RelationNet** | 62.45±0.98 | 63.11±0.94 | 68.65±0.91 | 67.59±1.02 | 66.20±0.99 |
| CUB 5-shot | **Baseline** | 64.16±0.71 | 73.07±0.71 | 81.27±0.57 | 82.85±0.55 | 84.27±0.53 |
|  | **Baseline++** | 79.34±0.61 | 82.02±0.55 | 85.17±0.50 | 83.58±0.54 | 84.50±0.51 |
|  | **MatchingNet** | 72.86±0.70 | 77.08±0.66 | 83.59±0.58 | 83.64±0.60 | 83.78±0.56 |
|  | **ProtoNet** | 70.77±0.69 | 78.14±0.67 | 84.76±0.52 | 87.42±0.48 | 85.98±0.53 |
|  | **MAML** | 72.09±0.76 | 76.31±0.74 | 80.33±0.70 | 82.70±0.65 | 83.47±0.59 |
|  | **RelationNet** | 76.11±0.69 | 77.81±0.66 | 81.12±0.63 | 82.75±0.58 | 82.30±0.58 |
| *mini*-ImageNet 1-shot | **Baseline** | 42.11±0.71 | 45.82±0.74 | 52.37±0.79 | 51.75±0.80 | 49.82±0.73 |
|  | **Baseline++** | 48.24±0.75 | 48.29±0.72 | 53.97±0.79 | 51.87±0.77 | 52.65±0.83 |
|  | **MatchingNet** | 48.14±0.78 | 50.47±0.86 | 54.49±0.81 | 52.91±0.88 | 53.20±0.78 |
|  | **ProtoNet** | 44.42±0.84 | 50.37±0.83 | 51.98±0.84 | 54.16±0.82 | 53.90±0.83 |
|  | **MAML** | 46.47±0.82 | 50.96±0.92 | 54.69±0.89 | 49.61±0.92 | 51.46±0.90 |
|  | **RelationNet** | 49.31±0.85 | 51.84±0.88 | 52.19±0.83 | 52.48±0.86 | 51.74±0.83 |
| *mini*-ImageNet 5-shot | **Baseline** | 62.53±0.69 | 66.42±0.67 | 74.69±0.64 | 74.27±0.63 | 73.45±0.65 |
|  | **Baseline++** | 66.43±0.63 | 68.09±0.69 | 75.90±0.61 | 75.68±0.63 | 76.16±0.63 |
|  | **MatchingNet** | 63.48±0.66 | 63.19±0.70 | 68.82±0.65 | 68.88±0.69 | 68.32±0.66 |
|  | **ProtoNet** | 64.24±0.72 | 67.33±0.67 | 72.64±0.64 | 73.68±0.65 | 74.65±0.64 |
|  | **MAML** | 62.71±0.71 | 66.09±0.71 | 66.62±0.83 | 65.72±0.77 | 65.90±0.79 |
|  | **RelationNet** | 66.60±0.69 | 64.55±0.70 | 70.20±0.66 | 69.83±0.68 | 69.61±0.67 |

## A8 MORE-WAY IN META-TESTING STAGE

We experiment with a practical setting that handles different testing scenarios. Specifically, we conduct the experiments of 5-way meta-training and N-way meta-testing (where N = 5, 10, 20) to examine the effect of testing scenarios that are different from training.

As in Table A6, we compare the methods Baseline, Baseline++, MatchingNet, ProtoNet, and RelationNet. Note that we are unable to apply the MAML method as MAML learns the initialization for the classifier and can thus only be updated to classify the same number of classes. Our results show that for classification with a larger N-way in the meta-testing stage, the proposed Baseline++ compares favorably against other methods in both shallow or deeper backbone settings.

We attribute the results to two reasons. First, to perform well in a larger N-way classification setting, one needs to further reduce the intra-class variation to avoid misclassification. Thus, Baseline++ has better performance than Baseline in both backbone settings. Second, as meta-learning algorithms were trained to perform 5-way classification in the meta-training stage, the performance of these algorithms may drop significantly when increasing the N-way in the meta-testing stage because the tasks of 10-way or 20-way classification are harder than that of 5-way one.

One may address this issue by performing a larger N-way classification in the meta-training stage (as suggested in Snell et al. (2017)). However, it may encounter the issue of memory constraint. For example, to perform a 20-way classification with 5 support images and 15 query images in each class, we need to fit a batch size of 400 (20 x (5 + 15)) that must fit into the GPUs. Without special hardware parallelization, the large batch size may prevent us from training models with deeper backbones such as ResNet.

Table A6: **5-way meta-training and N-way meta-testing experiment.** The experimental results are on mini-ImageNet with 5-shot. We could see Baseline++ compares favorably against other methods in both shallow or deeper backbone settings.

| | Conv-4 | | | ResNet-18 | | |
| N-way test | 5-way | 10-way | 20-way | 5-way | 10-way | 20-way |
|---|---|---|---|---|---|---|
| **Baseline** | 62.53±0.69 | 46.44±0.41 | 32.27±0.24 | 74.27±0.63 | 55.00±0.46 | 42.03±0.25 |
| **Baseline++** | 66.43±0.63 | **52.26±0.40** | **38.03±0.24** | **75.68±0.63** | **63.40±0.44** | **50.85±0.25** |
| **MatchingNet** | 63.48±0.66 | 47.61±0.44 | 33.97±0.24 | 68.88±0.69 | 52.27±0.46 | 36.78±0.25 |
| **ProtoNet** | 64.24±0.68 | 48.77±0.45 | 34.58±0.23 | 73.68±0.65 | 59.22±0.44 | 44.96±0.26 |
| **RelationNet** | **66.60±0.69** | 47.77±0.43 | 33.72±0.22 | 69.83±0.68 | 53.88±0.48 | 39.17±0.25 |

