# OpenReview forum: "A Closer Look at Few-shot Classification"
_ICLR.cc/2019/Conference_

### Official Review · AnonReviewer2 · 2018-10-24
**thought-provoking observations and nice comparative experiments**

**Rating:** 6
**Confidence:** 4

**Review:**

There are a few things I like about the paper.

Firstly, it makes interesting observations about the evaluation of the few-shot learning approaches, e.g. the underestimated baselines, and compares multiple methods in the same conditions. In fact, one of the reasons for accepting this paper would be to get a unified and, hopefully, well-written implementation of those methods.

Secondly, I like the domain shift experiments, but I have the following question. The description of the CUB  says that there is an overlap between CUB and ImageNet.  Is there an overlap between CUB and mini-ImageNet? If so, then domain shift experiments might be too optimistic or even then it is not a big deal?

One thing I don’t like is that, in my opinion,  the paper includes much redundant information which could go to the appendix in order to not weary the reader. For instance, everything related to Table 1. There is also some overlap between Section 2 and 3.3, while MAML, for instance, is still not well explained. Also, tables with too many numbers are difficult to read, e.g. Table 4.

---- Other notes -----

Many of the few-shot learning papers use Omniglot, so I think it would be a valuable addition to the appendix. Moreover, there exists a cross-domain scenario with Omniglot-> MNIST which I would also like to see in the appendix.

In the Matching Nets paper, there is a good baseline classifier based on k-NNs. Do you know how does that one compares to Baseline and Baseline++ models if used with the same architecture for the feature extractor?

The conclusion from the network depth experiments is that “gaps among different methods diminish as the backbone gets deeper”. However, in a 5-shot mini-ImageNet case, this is not what the plot shows. Quite the opposite: the gap increased. Did I misunderstand something? Could you please comment on that?

---

> ### Author Response · Authors · 2018-11-23
> **Responses to AnonReviewer2 -- part 2**
>
> Q4: In the Matching Nets paper, there is a good baseline classifier based on k-NNs. Do you know how does that one compares to Baseline and Baseline++ models if used with the same architecture for the feature extractor?
>
> A4: Here we show our 1-shot and 5-shot accuracy of Baseline and Baseline++ with the softmax and 1-NN classifier on the mini-ImageNet dataset with a Conv4 backbone. We only include the result of k = 1 with cosine distance to match the setting of Matching Nets paper.
>
> 1-shot
>  		      softmax		        1-NN (cosine distance)
> Baseline	      42.11% +- 0.71%	44.18% +- 0.69%
> Baseline++  48.24% +- 0.75%	49.57% +- 0.73%
>
> 5-shot
>  		      softmax		        1-NN (cosine distance)
> Baseline	      62.53% +- 0.69%	56.68% +- 0.67%
> Baseline++  66.43% +- 0.63%	61.93% +- 0.65%
>
> As shown above, using 1-NN classifier has better performance than that of using the softmax classifier in 1-shot setting, but softmax classifier is better in 5-shot setting instead. We note that that the number presented here are not directly comparable to the results reported in the Matching Nets paper because we use a different “mini-ImageNet” separation. In this paper, we follow the data split provided by [Ravi et al. ICLR 2017], which is used in most few-shot papers. We have included the result in the appendix of the revised paper.
>
> Q5: The conclusion from the network depth experiments is that “gaps among different methods diminish as the backbone gets deeper”. However, in a 5-shot mini-ImageNet case, this is not what the plot shows. Quite the opposite: the gap increased. Did I misunderstand something? Could you please comment on that?
>
> A5: Sorry for the confusion. As addressed in 4.3, gaps among different methods diminish as the backbone gets deeper *in the CUB dataset*. In the mini-ImageNet dataset, the results are more complicated due to the domain difference. We further discuss this phenomenon in Section 4.4 and 4.5. We have clarified related texts in the revised paper.

---

> ### Author Response · Authors · 2018-11-23
> **Responses to AnonReviewer2 -- part 1**
>
> Thanks for your opinions! Our responses are as follow:
> Q1:  Is there an overlap between CUB and mini-ImageNet? If so, then domain shift experiments might be too optimistic or even then it is not a big deal?
>
> A1:  There are only 3 out of 64 base classes that are *birds* in the mini-ImageNet dataset. Furthermore, these three categories (house_finch, robin, toucan) are different from the 200 bird categories in CUB. Thus, a large domain shift still exists between the mini-ImageNet and the CUB dataset.
>
> Q2: The paper includes much redundant information which could go to the appendix in order to not weary the reader. For instance, everything related to Table 1. There is also some overlap between Section 2 and 3.3, while MAML, for instance, is still not well explained. Also, tables with too many numbers are difficult to read, e.g. Table 4.
>
> A2: Thanks for the comments.
> First, our purpose for showing Table 1 is two-fold: 1) it validates our reimplementation by comparing results from the reported numbers and 2) it shows that the implementations of the Baseline method in prior works are underestimated.
>
> Second, we have included a more detailed description of MAML in the revised paper.
>
> Third, thanks for the suggestion. To improve the readability, we have modified Table 4 in the original paper to a figure (see Figure 5 in the revised paper). We include the detailed numbers in the appendix for reference.
>
> Q3:  Many of the few-shot learning papers use Omniglot, so I think it would be a valuable addition to the appendix. Moreover, there exists a cross-domain scenario with Omniglot-> MNIST which I would also like to see in the appendix.
>
> A3:  Thanks for the suggestions. We did not include Omniglot because its performance has been saturated in most of the recent work (~99%). We will add the results to the appendix in the camera-ready version for completeness. We agree that the Omniglot-> MNIST experiment will be a good addition to the paper. We will also add the results to the appendix in the camera-ready version.

---

### Official Review · AnonReviewer1 · 2018-10-31
**A nice experimental survey; experiment design could be improved**

**Rating:** 6
**Confidence:** 5

**Review:**

This paper gives a nice overview of existing works on few-shot learning. It groups them into some intuitive categories and meanwhile distills a common framework (Figure 2) employed by the methods. Moreover, the authors selected four of them, along with two baselines, to experimentally compare their performances under a cleaned experiment protocol.

The experiments cover three few-shot learning scenarios respectively for generic object recognition, fine-grained classification, and cross-domain adaptation. While I do *not* think the third scenario is “more practical”, it is certainly nice to have it included in the experiments.

The experiment setup is unfortunately questionable. Since there is a validation set, one should use it to determine the free parameters (e.g., the number of epochs, learning rates, etc.). However, it seems like the same set of free parameters are used for different methods, making the comparison unfair because this set may favor some methods and yet hurt the others.

The results of RelationNet are missing in Table 4.

Another concern is that the same number of novel classes is used in the training and the testing stage. A more practical application of the learned meta model is to use it to handle different testing scenarios. There could be five novel classes in one scenario, 10 novel classes in another, and 100 in the third, etc. The number of labeled examples per class may also vary from one testing scenario to anther.

It is misleading by the following: “Very recently, Motiian et al. (2017) addresses the few-shot domain adaptation problem.” There are a few variations in domain adaptation (DA). The learner has access to the fully labeled source domain and a small set of labeled target examples in supervised DA, to the source domain, a couple of labeled target examples, and many unlabeled target examples in semi-supervised DA, and to the source domain and many unlabeled target data points in the unsupervised DA. These have been studied long before (Motiian et al., 2017), for instance the works of Saenko et al. (2010) and Gong et al. (2013).

[ref] Saenko K, Kulis B, Fritz M, Darrell T. Adapting visual category models to new domains. InEuropean conference on computer vision 2010 Sep 5 (pp. 213-226). Springer, Berlin, Heidelberg.

[ref] Gong B, Grauman K, Sha F. Connecting the dots with landmarks: Discriminatively learning domain-invariant features for unsupervised domain adaptation. InInternational Conference on Machine Learning 2013 Feb 13 (pp. 222-230).

Overall, the paper is well written and may serve as a nice survey of existing works on few-shot learning. The unified experiment setup can facilitate the future research for fair comparisons, along with the three testing scenarios. However, I have some concerns as above about the experiment setups and hence also the conclusions.

---

> ### Author Response · Authors · 2018-11-23
> **Responses to AnonReviewer1  -- part 2**
>
> Q3: Another concern is that the same number of novel classes is used in the training and the testing stage. A more practical application of the learned meta model is to use it to handle different testing scenarios.
>
> A3: Thanks for pointing this out. As suggested, we conduct the experiments of 5-way meta-training and N-way meta-testing (where we vary the number of N to be 5, 10, and 20) to examine the effect of handling testing scenarios that are different from training. We compare the methods Baseline, Baseline++, MatchingNet, ProtoNet, and RelationNet. Note that we are unable to apply the MAML method as MAML learns the initialization for the classifier and can thus only be updated to classify the same number of classes.
>
> We show the experimental results on mini-ImageNet with 5-shot meta-training as follows.
>
> Backbone: Conv4
> 	                5-way test	                10-way test	                20-way test
> Baseline	        62.53% +- 0.69%	        46.44% +- 0.41%	        32.27% +- 0.24%
> Baseline++	66.43% +- 0.63%	        *52.26% +- 0.40%*	*38.03% +- 0.24%*
> MatchingNet	63.48% +- 0.66%	        47.61% +- 0.44%	         33.97% +- 0.24%
> ProtoNet	64.24% +- 0.68%	        48.77% +- 0.45%	        34.58% +- 0.23%
> RelationNet	*66.60% +- 0.69%*	47.77% +- 0.43%	        33.72% +- 0.22%
>
> Backbone: ResNet18
> 	                 5-way test	                10-way test	                 20-way test
> Baseline	         74.27% +- 0.63%	        55.00% +- 0.46%	         42.03% +- 0.25%
> Baseline++	 *75.68% +- 0.63%*	*63.40% +- 0.44%*	 *50.85% +- 0.25%*
> MatchingNet	 68.88% +- 0.69%	        52.27% +- 0.46%	         36.78% +- 0.25%
> ProtoNet	 73.68% +- 0.65%	        59.22% +- 0.44%	         44.96% +- 0.26%
> RelationNet	 69.83% +- 0.68%	        53.88% +- 0.48%	         39.17% +- 0.25%
>
> Our results show that for classification with a larger-way (e.g., 10 or 20-way) in the meta-testing stage, the proposed Baseline++ compares favorably against other methods in both shallow or deeper backbone settings.
>
> We attribute the results to two reasons.
> 1) To perform well in a larger N-way classification setting, one needs to further reduce the intra-class variation to avoid misclassification. Thus, in both shallow and deeper backbone settings, Baseline++ has better performance than Baseline.
>
> 2)  As meta-learning algorithms were trained to perform 5-way classification in the meta-training stage, the performance of these algorithms may drop significantly when increasing the N-way in the meta-testing stage because the tasks of 10-way or 20-way classification are harder than that of 5-way classification.
>
> One may address this issue by performing a larger N-way classification in the meta-training stage (as suggested in [Snell et al. NIPS 2017]). However, this may encounter the issue of memory constraint. For example, to perform a 20-way classification with 5 support images and 15 query images in each class, we need to fit a batch size of 400 (20 x (5 + 15)) that must fit in the GPUs. Without special hardware parallelization, the large batch size may prevent us from training models with deeper backbones such as ResNet. We have include the result in the appendix of the revised paper.
>
> Q4: It is misleading by the following: “Very recently, Motiian et al. (2017) addresses the few-shot domain adaptation problem.”...
>
> A4: Thanks for the correction. Indeed, both Saenko et al. Gong et al. address the supervised domain adaptation problem with only a few labeled instances prior to [Motiian et al., NIPS 2017].
>
> On the other hand, we would like to point out another research direction. Very recently, the method in [Dong et al. ECML-PKDD 2018] addresses the few-shot problem where both the domain *and* the categories change.  This work is more related to our setting, as we also consider novel category accuracy in few-shot classification under domain differences. We have corrected the statement in the revised paper.

---

> > ### Comment · AnonReviewer1 · 2018-11-27
> > **Regarding the third setting / Q4**
> >
> > I appreciate the authors' efforts in improving the experiments. Regarding the third setting (cross-domain adaptation), I still think it is not necessary to introduce it to few-shot learning, at least not now. Instead, it is probably better to focus on and try to advance the conventional problem setup for now. Moreover, as the authors point out, the third setting is related to several previously studied directions. I would recommend the authors to discuss those in the paper --- it is probably not a good idea to simply remove Motiian et al. (2017) in the revised PDF.

---

> > > ### Author Response · Authors · 2018-11-28
> > > **Reply about the third setting**
> > >
> > > Thanks R1 for the reply. Our goal of showing the cross-domain adaptation is to highlight the limitations of existing few-shot classification algorithms problem in handling domain shift. Our results in this setting show that 1) the baseline algorithm surprisingly outperforms all other few-shot classification methods and 2) the performance of few-shot classification algorithms can greatly benefit from further adaptation to the target domain even with a limited amount of data. We believe that our unified experimental setup will facilitate future efforts along this direction.
> > >
> > > In the following, we also provide a taxonomy of existing work in related topics based on the availability of labeled/unlabeled data in the target domain, we would add the table to the appendix of the camera ready version to provide a more complete picture for the readers.
> > >
> > > Domain adaptation (DA): Evaluated on the *same* classes
> > > 					                	                    Source domain		        Target domain
> > > 				                 Domain shift	Labeled 	Unlabeled	Labeled (few) Unlabeled
> > > Supervised DA
> > > [Saenko et al., ECCV 2010]	         V		              V	      -		                   V 		        -
> > > [Motiian et al., NIPS 2017]
> > >
> > >
> > > Semi-supervised DA		        V		              V	      -		                   V		        V
> > >
> > >
> > > Unsupervised DA 		        V		              V	      -		                    -		        V
> > >
> > >
> > >
> > >
> > > Few-shot classification: Evaluated on the *novel* classes
> > > 						                                          Base class		           Novel class
> > > 				                       Domain shift	Labeled 	 Unlabeled	Labeled (few) Unlabeled
> > > Few-shot			                         -		     V	                 -		           V		          -
> > >
> > >
> > > Cross-domain few-shot
> > > [Ours (third setting);		                V		     V	                 -		           V		          -
> > > Dong et al. ECML-KDD 2018]
> > >
> > >
> > > Semi-supervised few-shot
> > > [Ren et al. ICLR 18]		               -		     V	                 V		            V		          V

---

> > > > ### Comment · AnonReviewer1 · 2018-12-14
> > > > **Probably include the discussion in the paper**
> > > >
> > > > Thank the authors for the further response. The matrix of different settings seems informative. The authors are encouraged to include it in the paper.

---

> > > > > ### Author Response · Authors · 2018-12-16
> > > > > **Thank you, we will**
> > > > >
> > > > > Thank you, we will include them in the appendix of the revised manuscript.

---

> ### Author Response · Authors · 2018-11-23
> **Responses to AnonReviewer1 -- part 1**
>
> Thanks for your comments! Our responses are as follow:
> Q1: “Using validation set to determine the free parameters...”
>
> A1: Thank you for the comment. In our paper, we did use the validation set to select the best number of training iterations for meta-learning methods. Specifically, the exact iterations for experiments on the mini-ImageNet in the 5-shot setting with a four-layer ConvNet are:
>
> - ProtoNet:        24,600 iterations
> - MatchingNet: 35,300 iterations
> - RelationNet:   37,100 iterations
> - MAML:             36,700 iterations
>
> We have clarified this in the revised paper.
>
> On the other hand, we were not able to use the validation set for the Baseline and Baseline++. Note that validation set for few-shot problem splits by class, and does not split data in one class. With these validation classes in meta-training stage, one can validate how well the model can predict novel classes in meta-testing stage. However, the Baseline and Baseline++ methods cannot predict validation classes, as they has a fixed softmax layer to predict base classes. On the other hand, for meta-learning methods, the class to predict is conditioned on the class in the support set. Thus, with the support set in validation class, meta-learning methods can predict the validation class. As an alternative for Baseline and Baseline++, we directly train 400 epoches. We observe convergence from the training curve in both the Baseline and Baseline++ methods.
>
> For the learning rate and optimizer, we use Adam with an initial learning rate 0.001 for all of the methods because the ProtoNet, RelationNet, and MAML methods all use the same setting as described in the respective papers. However, we cannot find information about the learning rate for MatchingNet. The learning rate of 0.001 is also given as a default hyper-parameter for Tensorflow and PyTorch. The results in Table 1 of our paper ensure that the results reproduce the performance presented in the original papers.
>
> For other hyper-parameters such as the network depth in the backbone architecture, we have a detailed comparison as shown in Section 4.3 of the paper.
>
> Q2: The results of RelationNet are missing in Table 4.
>
> A2: Adapting RelationNet using training data in the support set (from novel classes) at the meta-testing stage is non-trivial. As the relation module in RelationNet takes convolution maps as input, we are not able to not replace the relation module with a softmax layer as we do for the ProtoNet and MatchingNet.
>
> As an alternative, at the meta-testing stage, we split the training data in the novel class into support and query data and use them to update the relation module. Specifically, we take the RelationNet with a ResNet-18 feature backbone. We randomly split the few training data in novel class into 3 support and 2 query data to finetune the relation module for 100 epochs. The results on CUB, mini-ImageNet and mini-ImageNet ->CUB are shown below.
>
> 		 CUB			mini-ImageNet	mini-ImageNet -> CUB
> original	 82.75% +- 0.58%	69.83% +- 0.68%	57.71% +- 0.73%
> adapted	 83.17% +- 0.57%	70.49% +- 0.68%	58.54% +- 0.72%
>
> In all three cases, adapting the relation module using the support data in the meta-testing stage improves the results. However, the improvement is somewhat marginal. We have included the additional results in the revised paper.

---

### Official Review · AnonReviewer3 · 2018-11-01
**Conclusion is a bit confusing**

**Rating:** 6
**Confidence:** 2

**Review:**

The paper tried to propose a systematic/consistent way for evaluating meta-learning algorithms. I believe this is a great direction of research as the meta-learning community is growing quickly. However, my question is if a relatively simple modification could improve the baselines, are there simple modifications available to other meta-learning algorithms being investigated? If the other algorithms are not as good as they claimed, can you give any insights on why and what to improve?

---

> ### Author Response · Authors · 2018-11-23
> **Responses to AnonReviewer3**
>
> Thanks for your comments! Our responses are as follow:
> Q1: If a relatively simple modification could improve the baselines, are there simple modifications available to other meta-learning algorithms being investigated?
>
> A1: The simple modification we made for the baseline approach is to replace the softmax layer with a distance-based classifier. However, among other meta-learning algorithms, only the MAML method is applicable to this modification. Both ProtoNet and MatchingNet already use distance-based classifier in their algorithm. RelationNet has its own relation module so is not applicable for this modification. While MAML could adopt this strategy, we did not include it into our experiment since our primary goal is not to improve one specific method.
>
> Q2:  If the other algorithms are not as good as they claimed, can you give any insights on why and what to improve?
>
> A2:
> Meta-learning for few-shot classification algorithms are not as good as they claimed because of the following two aspects:
>
> First, in the CUB setting, the gap among each algorithm diminished when using a deeper backbone. That is, with a deeper feature backbone, the improvement from different meta-learning algorithm become less significant. Our results suggest that both deeper backbones and meta-learning algorithms both aim to reduce intra-class variation for improving few-classification accuracy. Consequently, when intra-class variation has been dramatically reduced using a deeper backbone, the contribution from meta-learning becomes less significant.
>
> Second, in the CUB -> mini-ImageNet setting where a larger domain shift exists, the Baseline method outperforms all meta-learning algorithms. That is, existing meta-learning algorithms are not robust to larger domain shift. As discussed in section 4.4, while meta-learning methods learn to learn from the support set during the meta-training stage, all of the base support sets are still within the same dataset. Thus, these algorithms did not learn how to learn from a support set with large domain shift.
>
> With our results, we encourage the community to tackle the challenge of potential domain shifts in the context of few-shot learning. We will release the source code and evaluation setting that will facilitate future research directions.

---

### Public Comment · (anonymous) · 2018-12-17
**Experiments**

Hi,

This is a good insight for different backbones impacting the performance in few-shot classification.

I want to verify several things here.

1. Did the authors run your learning  for matching networks, prototypical networks, maml, and relation networks with episodic training (sampled from N-classes and K-Shot every episode) from plain networks(conv and resnet)? Or did you train from baseline networks(pre-trained)?
2. What is the number of iteration here? is it the number of episode? or the number you learn the feature extractor (baseline)?
3. In MAML paper, they stated that  using 64 filters may cause overfitting, do the authors suffer the same thing as you change the backbone of MAML?

Thanks in advance.

---

> ### Author Response · Authors · 2018-12-18
> **Response to the three questions**
>
> Hi, thanks for your questions! We reply to the three questions below.
>
> 1. Did the authors run your learning for matching networks, prototypical networks, maml, and relation networks with episodic training (sampled from N-classes and K-Shot every episode) from plain networks(conv and resnet)? Or did you train from baseline networks(pre-trained)?
>
> Yes, we train all the networks (including matching networks, prototypical networks, MAML, and relation networks) with episodic training from the plain networks. All the networks were randomly initialized with He initialization, the standard initialization used in ResNet.
>
> 2. What is the number of iteration here? is it the number of episode? or the number you learn the feature extractor (baseline)?
>
> Yes, the number of iterations refers to the number of the episode. Thanks for pointing this out. We will use the number of episode in the revised manuscript for clarity.
>
> 3. In MAML paper, they stated that using 64 filters may cause overfitting, do the authors suffer the same thing as you change the backbone of MAML?
>
> We do not see the overfitting effect from observing the validation loss. We believe that it is due to the data augmentation used in all our experiments.

---

> > ### Public Comment · (anonymous) · 2018-12-21
> > **Thank you**
> >
> > Thank you for the answers.
> > I do appreciate this work. It provides rigorous experiments.

---

### Public Comment · (anonymous) · 2018-12-22
**Reproduce**

Hi authors,

I want to reproduce the results on various methods (MatchingNets, ProtoNets, etc.) with ResNet backbones here.
But I could not get the performance like in the paper.
Would you mind to give hyper parameters and image size detail? or tricks that you used?

---

> ### Author Response · Authors · 2018-12-23
> **Response**
>
> Thanks for your questions.
>
> For hyper-parameters, we use the standard parameters of the ResNet architecture. We use the standard image size of 224 x 224 as input to the ResNet. We believe that the major difference between our re-implementation and the publicly available code for meta-learning methods (including ProtoNet, MatchingNet, MAML, and relation networks) lies in the data augmentation.
>
> We expect to release our code by mid-Jan. Before that, we would be happy to chat more if you have specific questions. Feel free to drop us an email at weiyuc@andrew.cmu.edu

---

> > ### Public Comment · (anonymous) · 2019-01-31
> > **Avaliable Code**
> >
> > Hi, thanks for this interesting work. Do you have public available codes to reproduce the results?

---

> > > ### Author Response · Authors · 2019-02-03
> > > **Code publicly available**
> > >
> > > Thank you for your interests. We have made our code publicly available: https://github.com/wyharveychen/CloserLookFewShot
> > >
> > > Please free to drop us an email at weiyuc@andrew.cmu.edu if you have any questions.

---

### Public Comment · (anonymous) · 2019-07-25
**Fair Comparison?**

The baseline and baseline++ model use all available examples in each class to train instead of n shots, while the meta learning methods only use n shots to train. In other words, the baseline/baseline++ methods see more data points in training. I wonder such comparison is fair.

---

> ### Author Response · Authors · 2019-07-25
> **Response**
>
> Thanks for your comment. However, both meta-learning methods and baseline/baseline++ methods see all available data points in training. Despite meta-learning methods only use n shots to train in each episode, the n shot batches they use are different in each episode.

---

> > ### Public Comment · (anonymous) · 2019-07-30
> > **Response**
> >
> > Thanks for the response. Now it makes sense to me. Although meta-learning might not see every data point through random sampling, but it should see most of them.

---

### Meta-Review · Area_Chair1 · 2018-12-13
**An intriguing experimental paper on the current state of few-shot learning.**

**Confidence:** 4
**Recommendation:** Accept (Poster)

**Metareview:**

This paper provides a number of interesting experiments for few-shot learning using the CUB and miniImagenet datasets. One of the especially intriguing experiments is the analysis of backbone depth in the architecture, as it relates to few-shot performance. The strong performance of the baseline and baseline++ are quite surprising. Overall the reviewers agree that this paper raises a number of questions about current few-shot learning approaches, especially how they relate to architecture and dataset characteristics.

A few minor comments:
- In table 1, matching nets are mistakenly attributed to Ravi and Larochelle. Should be Vinyals et al.
- The notation for cosine similarity in section 3.2 is odd. It looks like you’re computing some cosine function of two vectors which doesn’t make sense. Please clarify this.
- There are a few results that were promised after the revision deadline, please be sure to include these in the final draft.